# A distributed temperature profiling method for assessing spatial variability of ground temperatures in a discontinuous permafrost region of Alaska

Emmanuel Léger[1], Baptiste Dafflon[1], Yves Robert[1], Craig Ulrich[1], John E. Peterson[1], Sébastien Biraud[1], Vladimir E. Romanovsky[2], and Susan S. Hubbard[1]

[1]Lawrence Berkeley National Laboratory, Berkeley, CA, 94720, USA
[2]Geophysical Institute, University of Alaska Fairbanks, Fairbanks, 99775, USA

*Correspondence to*: Baptiste Dafflon (bdafflon@lbl.gov)

**Abstract.** Soil temperature has been recognized as a property that strongly influences a myriad of hydro-biogeochemical processes, as well as reflecting how various properties modulate the soil thermal flux. In spite of its importance, our ability to acquire soil temperature data with high spatial and temporal resolution and coverage is limited, because of the high cost of equipment, the difficulties of deployment, and the complexities of data management. Here we propose a new strategy that we call Distributed Temperature Profiling (DTP), for improving the characterization and monitoring of near-surface thermal properties through the use of an unprecedented number of laterally and vertically distributed temperature measurements. We developed a prototype DTP system, which consists of inexpensive, low-impact, low-power, vertically resolved temperature probes that independently and autonomously record soil temperature. The DTP system concept was tested by moving sequentially the system across the landscape, to identify near-surface permafrost distribution in a discontinuous permafrost environment near Nome, Alaska during the summer time. Results show that the DTP system enabled successful acquisition of vertically resolved profiles of summer soil temperature over the top 0.8 m at numerous locations. DTP also enabled high resolution identification and lateral delineation of near-surface permafrost locations from surrounding zones with no permafrost or deep permafrost table locations overlain by a perennially thawed layer. The DTP strategy overcomes some of the limitations associated with —and complements the strengths of— borehole-based soil temperature sensing as well as Fiber-Optic Distributed Temperature Sensing (FO-DTS) approaches. Combining DTP data with co-located topographic and vegetation maps obtained using Unmanned Aerial Vehicle (UAV) and Electrical Resistivity Tomography (ERT) data allowed us to identify correspondences between surface and subsurface property distribution, and in particular between topography, vegetation, shallow soil properties, and near-surface permafrost. Finally, the results highlight the considerable value of the newly developed DTP strategy for investigating the significant variability and complexity of subsurface thermal and hydrological regimes in discontinuous permafrost regions.

## 1 Introduction

Soil temperature and its spatial and temporal variability mediate a myriad of above- and belowground hydro-biogeochemical processes. Soil temperature is an important factor influencing the water and energy exchanges with the atmosphere, including evaporation (Smits et al., 2011). In addition, all chemical and biochemical reactions in soil, including those related to root and soil respiration and microbial decomposition, are temperature dependent (Davidson and Janssens, 2006). Thus, soil temperature plays an important role in plant growth and in soil carbon efflux and feedback on atmospheric $CO_2$ (Fang and Moncrieff, 2001).

Soil temperature influences many processes, but in turn it is controlled by climatic forcing and modulated by canopy characteristics, snow insulation, surface water, soil thermal parameters, and heat and water fluxes in the subsurface. Therefore, time-series of soil temperature can be used to estimate the influence of the above factors on the thermal regime. For example, time-series of temperature measurements can be used in a parameter estimation framework to quantify the thermal parameters and, potentially, the fraction of soil constituents including organic matter content (Nicolsky et al., 2009; Tran et al., 2017). Thermal temporal variability is also used to investigate fluid fluxes, surface water/groundwater exchange, and groundwater recharge (Briggs et al., 2012; Stonestrom and Constantz, 2003).

The significant spatial and temporal variability in the aforementioned processes require surveying and/or monitoring multiple locations to capture and understand the heterogeneity of the studied system. Conventional point-sensor methods for characterizing and monitoring soil temperature predominantly rely on measurements collected using point sensors placed directly in the ground, or deployed as a string of sensors along a probe or a borehole. Different types of sensors are commonly used, including thermistors, thermocouples, and temperature-sensing integrated circuits (Mukhopadhyay, 2013). While sensors with analog output have been widely used for temperature measurements requiring high resolution and accuracy, sensors with digital output are improving continuously and offer a promising alternative for numerous applications.

Usually, a data logger is physically connected to multiple thermal sensors, although a growing number of studies use self-recording sensors that collect and store the data individually (including iButtons, Onset Pendants, LogTag, UTL-3) (Gisnås et al., 2014; Hubbart et al., 2005; Lundquist and Lott, 2008) to increase the number of spatially distributed temperature measurements at a reasonable cost. Fiber-Optic Distributed Temperature Sensing (FO-DTS) offers an alternative to point-sensor methods in studies where temperature measurements with high spatial and temporal-sampling resolution are needed (Tyler et al., 2009). The optimal sensing approach is case specific and depends on many factors and requirements, including material, deployment and management costs, spatial and temporal resolution and coverage, data resolution, and data accuracy (e.g., Lundquist and Lott, 2008).

While the cost per traditional temperature point sensor can be considered low (in the range of $1 to $150), the total cost using the point-sensor method—including data logger, packaging, installation, localization and management—make this method often expensive to install in large numbers. Several studies have focused on evaluating various approaches to

decrease the cost and increase the number of measurement locations. The best example of this was the deployment of ~1600 self-recording temperature sensors (TRIX-16 Logtag sensors) across a domain extending from the Boise Basin, Idaho, to southern British Columbia, to evaluate downscaling of air temperature from long-term weather stations—using covariates that had established physical links to surface air temperature, including solar insolation, soil moisture, local topography,

canopy cover, geopotential height, and humidity (Holden et al., 2016). In another study, 390 self-recording temperature sensors (iButtons) were deployed to record the distribution of ground-surface temperature in a region with high topographic variability in the Swiss Alps. The acquired data were used to document the effect of elevation, slope, aspect and ground cover type on the mean annual ground-surface temperature (Gubler et al., 2011). Similarly, 171 sensors (mostly iButtons) recording the distribution of ground surface temperatures across a climatic gradient from continuous to sporadic permafrost

in Norway documented the pronounced control of snow depth on the local-scale variability of mean annual ground-surface temperature (Gisnås et al., 2014). While networks of low-cost distributed temperature sensors have concentrated primarily on air temperature measurements (Alcoforado and Andrade, 2006; Holden et al., 2016; Hubbart et al., 2005; Whiteman et al., 2000) and ground-surface temperature (Davesne et al., 2017; Gisnås et al., 2014; Gubler et al., 2011; Lewkowicz et al., 2012; Lundquist and Lott, 2008), little effort has been made to increase vertical- and lateral-direction temperature

measurements in soil. Important exceptions include the measurement of active layer thickness and/or soil temperature at multiple locations across 10x10 m to 1000x1000 m areas at sites as part of the Circumpolar Active Layer Monitoring (CALM) program (Nelson et al., 1998; Shiklomanov et al., 2008) and other sites (e.g., Goyanes et al., 2014; Guglielmin, 2006). Besides these efforts, the installation of sensor networks for soil temperature has typically been too spatially sparse to identify local-scale vertical and lateral variations in soil thermal regimes. Such fine-scale variations are relevant for

optimally quantifying, (among other effects) the influence of soil-snow-inundation-topographic-vegetation properties on the subsurface thermal regime, the fraction of soil constituents at numerous locations, and the role of subsurface hydrology and advective heat transport in permafrost distribution and evolution.

Note that while the development of Fiber-Optic Distributed Temperature Sensing (FO-DTS) has offered some promise in providing soil temperature measurements with high spatial and temporal-sampling resolution, this approach is limited to

specific applications and requires significant initial investment (>$30K, Lundquist and Lott, 2008), as well as careful experimental design to produce to its capacity (Lundquist and Lott, 2008; Tyler et al., 2009). In particular, FO-DTS deployment can require a dynamic calibration (Hausner et al., 2011), the occasional need of a fusion splicer to join fibers in the field, the disturbance of the investigated environment by the creation of a trench or a crack while installing and removing the cable, and the risk of losing a large amount of data in case of instrument, cable or power failure. FO-DTS is primarily

well suited where these issues can be easily addressed, such as for applications in a streambed, at the ground surface, in wells, in trenches, and in artificial ecosystems (Briggs et al., 2012).

Quantifying soil temperature has shown to be particularly important for understanding the evolution of permafrost in Arctic, sub-Arctic, Antarctic, and cold mountainous regions (e.g., Brewer, 1958; Guglielmin, 2006; Harris et al., 2001; Isaksen et al., 2011; Jorgenson et al., 2010; Lachenbruch and Marshall, 1969; Shiklomanov et al., 2008). In the Arctic,

Brewer (1958) recognized the dramatic influence of surface hydrology on permafrost thawing by using thermistor strings (Swartz, 1954), which monitored temperature in a lake down to depths of a few tens of meters below the lake bottom near Barrow, Alaska, over the course of a year. Lachenbruch and Marshall (1969) subsequently studied the effect of latent heat on permafrost temperature near shorelines and lakes where thermal profile anomalies were observed. In the 1980s, several studies focused on Arctic permafrost, including its relationship to historical temperature and climate change trends (Osterkamp, 1987, 1983, 1985; Osterkamp and Gosink, 1991). Osterkamp (1985) improved permafrost temperature measurements by developing a long thermistor cable that could sense temperature with high precision at its end. Many studies investigated further permafrost thermal hydrology and long-term temperature variations (e.g., Biskaborn et al., 2015; Burn, 2002; Osterkamp, 1987; Romanovsky and Osterkamp, 2000). Besides the use of point sensors for temperature measurements, FO-DTS has been applied in a few cases, including monitoring permafrost temperature along a transportation infrastructure (Roger et al., 2015) and detecting permafrost degradation during a controlled warming experiment (Wagner et al., 2018). In both cases, possible long-term disturbance resulting from the FO-DTS installation was not addressed, because the installation was made in the context of infrastructure improvement in the first case and short-term experiment in the second case.

Studies conducted over the last six decades in the Arctic have led to a steady improvement in our ability to evaluate permafrost distribution and characteristics, as well as our ability to evaluate the complex influence of various soil, vegetation, and atmosphere factors. These factors include snow cover (Stieglitz et al., 2003; Zhang, 2005), air temperature (Zhang et al., 1996), vegetative layers (Sturm et al., 2001), soil thermal parameters (Romanovsky and Osterkamp, 1995; Tran et al., 2017), soil hydrological properties (Dafflon et al., 2017), $CO_2$ and methane fluxes (Wainwright et al., 2015), and geomorphology (Jorgenson et al., 2010; Rowland et al., 2011). At local scales, studies have also shown the complexity of the system and its improved understanding once integrating multiple approaches, including soil sampling, point sensor methods, geophysical techniques, and remote sensing (e.g., Dafflon et al., 2016; Goyanes et al., 2014; Hubbard et al., 2013).

Despite many advances in understanding the Arctic Ecosystem functioning, improving the acquisition of spatially and temporally dense soil temperature measurements over relevant spatial scales is still critically important for advancing the predictive understanding of natural and managed ecosystems. Improving our predictive understanding of the interaction between plant distribution/dynamic and subsurface thermal and hydro-biogeochemical processes requires spatially and temporally dense measurements that yield important information about the energy and water fluxes in the subsurface. Indeed, the energy exchange at the ground surface and the heat flux in the subsurface are strongly mediated by snow, surface water, vegetation, and soil thermal properties, including peat layer thickness (Cable et al., 2016; Jorgenson et al., 2010), with each of these factors being highly spatially and temporally variable. Although this complexity has been recognized, the ability to quantify each of these processes and how they influence soil thermal and hydro-biogeochemical processes over time is still limited. Improving our ability to quantify how the lateral and vertical heterogeneity of processes are related is necessary, both to advance our mechanistic understanding, and to develop multiscale sensing and modeling strategies that better simulate hydro-biogeochemical processes and ecosystem evolution in a changing climate.

In this study, we introduce a novel sensing strategy that we call Distributed Temperature Profiling (DTP)—a strategy for obtaining spatially and temporally dense soil temperature measurements at flexible spatial scales—and then we test this strategy to investigate the permafrost distribution in a discontinuous permafrost environment. To this end, we built a prototype DTP system that consists of low-cost, low-impact, independent, vertically resolved temperature probes. Low-cost is defined here as being possibly built at a cost of less than $100 per probe and logger, not requiring any annual fee, and deployable at hundreds to thousands of locations. This approach fully explores the development of inexpensive, nimble, and low-powered single-board computers coupled with the large variety of sensors available owing to the development of the "Internet of Things" (Ashton, 2009) and "Makers Movements" (Dougherty, 2012). The DTP prototype provides measurements over the first meter below ground surface at numerous locations. We tested the DTP strategy to investigate the local distribution of near-surface thermal properties and the associated interpretation of near surface permafrost, and its link with surface properties. The study was performed within a 125 x 350 $m^2$ area in a discontinuous permafrost environment by moving sequentially the system across the landscape. Here we define near-surface permafrost by the relative absence of a year-round unfrozen soil layer (i.e., the absence of a Talik) above the permafrost table. We surveyed more than one hundred locations on July 17, 2017 —with repeated measurements at 40 locations on September 20, 2017. We further compared DTP measurements with Electrical Resistivity Tomography (ERT) and Unmanned Aerial Vehicle (UAV) data collected at the site around July 17, 2017, to document the value of the DTP measurements for interpreting permafrost variability and possible controls.

**2 Site Description**

We performed our study in a watershed about 40 km northwest of Nome, Alaska, specifically along Teller Road, as part of the Next Generation Ecosystem Experiment project (NGEE-Arctic) (Figure 1). This watershed can be considered to be representative of discontinuous permafrost systems, based on our preliminary investigations at the site, a numerical study evaluating the role of preferential snow accumulation in through Talik development under similar meteorological forcing (Jafarov et al., 2018) and other studies performed on the South Seward Peninsula (e.g., Yoshikawa and Hinzman, 2003). This study is the first (to our knowledge) to evaluate the permafrost distribution and co-variability with surface properties at this site. The watershed is characterized by a 130 m elevation gradient, the presence of solifluction lobes, a stream with a few confluents, and a diverse vegetation cover—including tall shrubs, dwarf shrubs, mosses, and graminoids (Figure 1). The geology across the watershed is defined by quaternary deposits recovering the Devonian to Ordovician geological unit (DOx unit), which is composed of mixed marble, schist, and graphitic metasiliceous rock (Hopkins and Karlstrom, 1955; Till et al., 1986). An outcrop at the bottom of the watershed along the main stream reveals schist, which is likely part of the mixed schist and marble sequence in the upper part of Dox (Till et al., 1986). Based on visual observations, the bedrock is overlaid, at least at some locations, by unconsolidated glacial and/or fluvial deposits of sand, gravel, cobble and boulder, with some visible at the surface. The thickness of the soil layer recovering the glacial/fluvial deposits and/or to bedrock is likely very

heterogeneous across the watershed, possibly varying from centimetres to several meters. The soil layer consists in an organic rich upper part and shows a gradual increase in bulk density with depth. A preliminary analysis of five soil samples collected in the 0.1-0.2 m depth interval at various locations shows dry bulk density ranging between 0.13 and 0.28 g/cm$^3$ and organic matter density ranging between 0.13 and 0.18 g/cm$^3$. Five other soil samples located in the 0.25-0.5 m depth

interval show dry bulk density ranging between 0.64 and 2.12 g/cm$^3$ and organic matter density ranging between 0.07 and 0.14 g/cm$^3$. The National Ocean and Atmospheric Administration's (NOAA's) meteorological station at Nome Municipal Airport indicates that over a five-year average (2013 to 2017) the mean annual air temperature is -1.02°C, the yearly rain precipitation is 450.6 mm, and the yearly snow fall is 1704.8 mm. Across the investigated watershed, the snow depth varies significantly from about 0.2 to 2 m, depending on the location and the year.

The main DTP survey was conducted along five 120 m long transects within a 125 x 350 m$^2$ study area (Figure 1) on July 17, 2017, during a period expected to be near or at the peak of the vegetation growing season. During that campaign, sparse measurements collected with a 1 m tile probe indicated that the thawed layer is most frequently thicker than one meter, although thinner at several locations with a minimum of 0.4-0.6 m thick at a few locations. Also, note that at locations where the tile probe encountered resistance between 0.6 and 1 m, the distinction between permafrost and the presence of rocky soil

is not always possible. A second short campaign, in which only DTP data were acquired along two of the five transects, took place on September 20, 2017 at the end of the summer season.

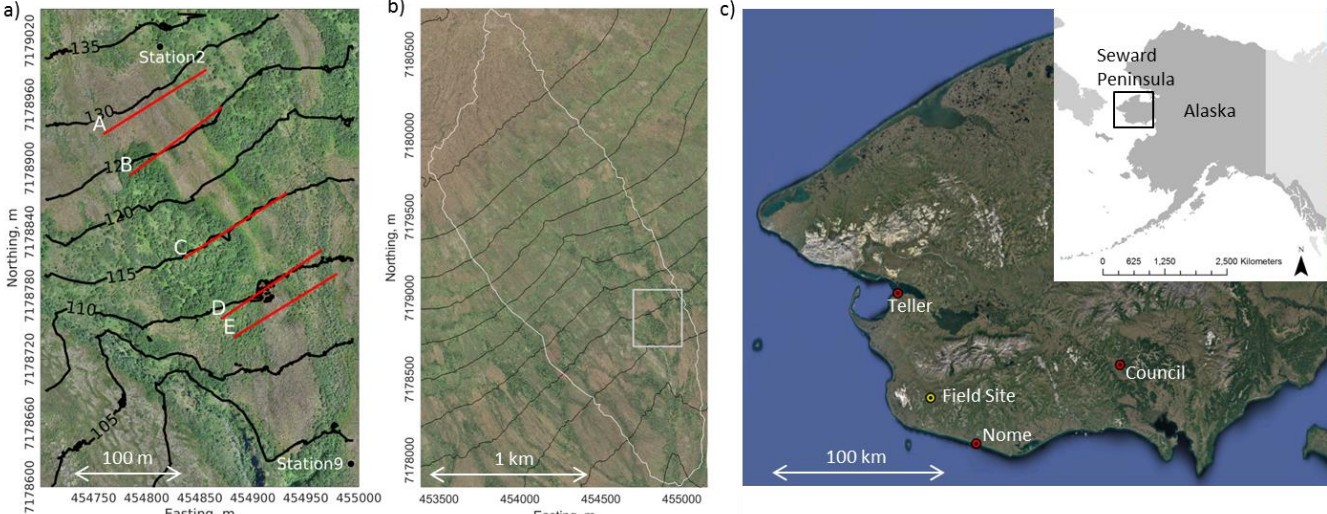

**Figure 1: Location and general setting of the study area. a) Aerial view of the investigated site, which includes a hillslope and a flatter toe area. Tall shrubs are dark green, mosses are bright green, and graminoids and dwarf-shrub dominated areas are light**
**brown. The RGB-mosaic is overlain with the location of the two long-term monitoring stations in this area, the location of transects surveyed in this study, and topographic isoline for every 5 m of elevation (in meter amsl.). b) Location of the investigated site (grey rectangle) in the Teller watershed. Topographic isoline for every 20 m of elevation. c) Location of the investigated field site on the South Seward Peninsula in Alaska.**

## 3 Materials and Methods

### 3.1 Distributed Temperature Profiling Strategy

The fundamental concept behind the DTP system involves using a network of vertically resolved temperature probes and accompanying loggers that provide temperature at multiple depths and locations, and enables deployment over tens to thousands of locations because of its low cost and automated data acquisition and management. Note that such a system can be deployed both as a characterization tool or for monitoring purposes. This study concentrates on the use of this strategy for characterizing soil temperature at numerous locations by moving the DTP system sequentially across the landscape. A DTP prototype system involving 30 probes was designed and built at the Lawrence Berkeley National Laboratory. While the number of probes is still rather small, to our knowledge this is the first time that such a vertically and laterally dense survey of soil temperature has been realized.

Each probe included 11 digital thermometers located 8 cm apart in the vertical direction. Each sensor was soldered on a thin copper sheet, inserted in a 9.5 mm outside-diameter PVC tube, and thermally isolated from other digital thermometers on the probe by epoxy-based glue. The digital thermometers used were the DS18B20 (Maxim Integrated$^{TM}$), which were 12 bits corresponding to a resolution of 0.0625°C and sold by the manufacturer as ±0.5°C maximum error (https://datasheets.maximintegrated.com/en/ds/DS18B20.pdf). Data logging was performed individually for each probe across the network using a coupled Raspberry-Pi 3 single-board computer and a PYTHON-based acquisition protocol. The material involved in the construction of each probe with its coupled logger cost ~US $90 (including the Raspberry-Pi 3). The probe sleeve and filling material were partly influenced by the work of Bill Cable, who built probes with vertically placed, highly accurate analog thermistors (led by Bill Cable, UAF, Alaska, http://permafrost.gi.alaska.edu/content/thermistor-probe-construction). While Cable's work was relevant for obtaining high vertical resolution at specific monitoring locations, here, this study is a first step toward building a highly duplicable probe and logger integrated system. Importantly, the prototype described here was intended for testing an acquisition strategy, and in no way represented the ideal DTP system with regard to hardware and software. Based on the results of this study, research is in progress to develop a DTP system with an extraordinarily low production and assembly cost; miniaturized data-logger; automated data acquisition, management, and transfer; and open source software and hardware to encourage community-based development and deployment.

The DTP prototype system was deployed sequentially at several locations in the watershed. A tile probe with the same diameter as the temperature probe was first used to create a hole in the soil, wherein the temperature probe was then inserted while still being in tight contact with soil. The 80 cm tall probes were inserted into the ground every 5 m along each transect, and left in place for data acquisition for ~30 minutes to ensure thermal equilibrium with the soil temperature. They were then moved to the next position. Thirty minutes to ensure thermal equilibrium was defined as a safe choice, based on preliminary tests that showed that 20 minutes were needed to approach a constant temperature in a +/-0.1°C range. Also, the influence of soil temperature fluctuation that may occur during the day was evaluated using two probes that monitored soil temperature.

This dataset showed that only the shallowest temperature sensors (≈ top 25 cm of soil) were affected by variations in atmospheric forcing during the day of the survey. The fluctuation interval was less than 5°C at 0 cm depth, 3°C at 8 cm depth and 1°C at 16 cm depth. Although obtained at only a few locations, these observations provide a rough indication of the existing limitation in comparing shallow soil temperatures from various locations measured at different time of the day.

Processing of the acquired temperature data has been minimal. To ensure the comparison of soil temperature at the same depth at various locations, we interpolated the measurement at specific depths. We also conducted a linear fit of the three deepest measurements along the probe to extrapolate temperature to 0.8 m depth. Extrapolation of temperature to 0.8 m depth occurred only at five locations along the transect E where the probes could not be pushed down to 0.8 m depth, due to the presence of permafrost or rock. Further, the DTP measurements presented here did not get corrected with any in-house

calibration factors. A calibration bath at temperature close to 0°C showed that the error was better than the +- 0.5°C maximum error indicated by the manufacturer, with all the measurements related to an error less than +- 0.25°C. We decided to not infer a sensor-specific calibration curve to improve the sensor accuracy because of several sources of uncertainty identified in our calibration approach including suboptimal calibration set up for such long probes, the limited intrinsic resolution of the sensor, and the limitation in obtaining highly accurate reference measurements.

**3.2 Point-scale Measurements, Including Temperature at Monitoring Stations**

All measurement locations and elevations were surveyed with a Real-Time Kinematic (RTK) GPS, with centimeter accuracy in latitude/longitude positioning and elevation. Average soil water content in the upper 30 cm of soil was estimated at each DTP probe location along the transects using a Time Domain Reflectometry (TDR) probe (6050X1 TRASE System I portable unit) with 30 cm metallic probes. In addition to the DTP data, soil temperature data were also obtained from long-

term conventional thermal monitoring stations established in the watershed, each of which included five conventional temperature sensors and a data logger (Onset, Cape Cod, Massachusetts). The reported accuracy of these temperature sensors is 0.25°C; however, an ice bath calibration was performed prior to installation, improving the accuracy for temperatures near 0°C to approximately 0.03°C (Cable et al., 2016). Each of these conventional temperature sensors was taped to a 5 mm diameter PVC rod vertically inserted to different depths, including 0.02, 0.25, 0.5, 1 and 1.5 m below ground surface.

Monitoring Stations 2 and 9, which were close to the investigated zone, were used to evaluate year-round temporal changes in temperature.

**3.3 Electrical Resistivity Tomography**

Electrical Resistivity Tomography data are typically collected using electrodes inserted into the ground, where the current is injected between two electrodes and the electrical potential difference is determined between two others (Binley and Kemna,

2005). The acquired resistance dataset is then inverted to estimate the spatial distribution of soil electrical resistivity (Rücker et al., 2017). The electrical conductivity (or its inverse, electrical resistivity) response is influenced by subsurface properties such as water/ice content, fluid electrical conductivity, lithological properties such as clay content, and soil temperature

(Schön, 2015). ERT is increasingly used to identify permafrost distribution and characteristics, and to complement other measurements, including soil temperature (e.g., Dafflon et al., 2016; Léger et al., 2017; Minsley et al., 2012). In this study, ERT was used to complement DTP data, and in particular the assessment of the potential links between what is observed in the top meter of soil and the deeper subsurface heterogeneity in physical, hydrological and thermal properties. Given that advanced analysis and interpretation of ERT data is beyond the scope of this study, here we qualitatively compared DTP and ERT signatures and discuss their joint value for inferring the presence of near-surface permafrost.

The electrical resistivity surveys were carried out using a MPT DAS-1 system with a 120-electrode structure and one-meter spacing. The data were acquired in the frequency domain using dipole-dipole geometry. ERT data were inverted using the Boundless Electrical Resistivity Tomography (BERT) code (Rücker et al., 2006; Rücker et al., 2017; Rücker and Spitzer, 2006), which is a finite-element-based inversion process. No temperature correction was applied to the inverted ERT data, because of the large range of resistivity values observed compared to the effect of temperature on the data, and because of the unavailability of spatially distributed temperature measurements deeper than 0.8 m depth.

Based on permafrost resistivity ranges associated with field datasets (e.g., Dafflon et al., 2016; Hilbich et al., 2008; Krautblatter et al., 2010; Marescot et al., 2008) and laboratory datasets (Hauck, 2002; Wu et al., 2013), and assuming low salinity and low clay content at the investigated site, we postulated that high resistivity values from 1000 to 7000 Ohm.m were primarily related to the presence of permafrost when encountered close to the surface (in the top 4 meters). Those high values may also be related to the presence of bedrock or permafrost if encountered deeper below the surface. Based on Dafflon et al. (2017) at an Arctic site in Utqiagvik, Alaska, resistivity values below 400 Ohm.m were interpreted to correspond to unfrozen conditions, while values between about 400 and 1000 Ohm.m were interpreted to correspond to frozen, partially frozen, or unfrozen conditions.

### 3.4 Unmanned Aerial Vehicle

UAV-based optical imagery was collected to reconstruct a color orthomosaic and Digital Surface Model (DSM) in order to understand vegetation distribution and topography. UAV-based imaging was performed during the July 2017 campaign using a 3DR Solo UAV and a Sony 5100 as a sensor. The orthomosaic and DSM (Figure 1) were reconstructed using a commercial software (PhotoScan from Agisoft LLC) and georeferenced using targets set on the ground and measured with a RTK GPS (with a workflow similar to Dafflon et al., 2016). The final resolution and uncertainty of the DSM and color orthomosaic were about 4 cm in x,y,z directions. To estimate a digital terrain model (DTM) proxy from the DSM, we re-interpolated the elevation after removing pixels showing a difference greater than 0.5 m between their elevation and the minimum elevation in a centered 10x10 $m^2$ window. This enabled us to partially remove the presence of shrubs, while the obtained DTM proxy involves variable spatial resolution that is always lower than the original DSM.

### 4 Results

Figure 2 shows the soil temperature at 0.8 m depth on July 17 obtained from the DTP system overlain on the reconstructed color orthomosaic. The DTP dataset shows that the lateral variability in soil temperature at 0.8 m depth is very low over some distance intervals but very abruptly varying in several other locations, with changes of up to 6°C occurring over a 5 m distance or less (Figure 2). Most abrupt lateral changes in temperature at 0.8 m depth occur at 15 and 75 m along transect A, at 40 and 75 m along transect B, and at 35 and 65 m along transect E.

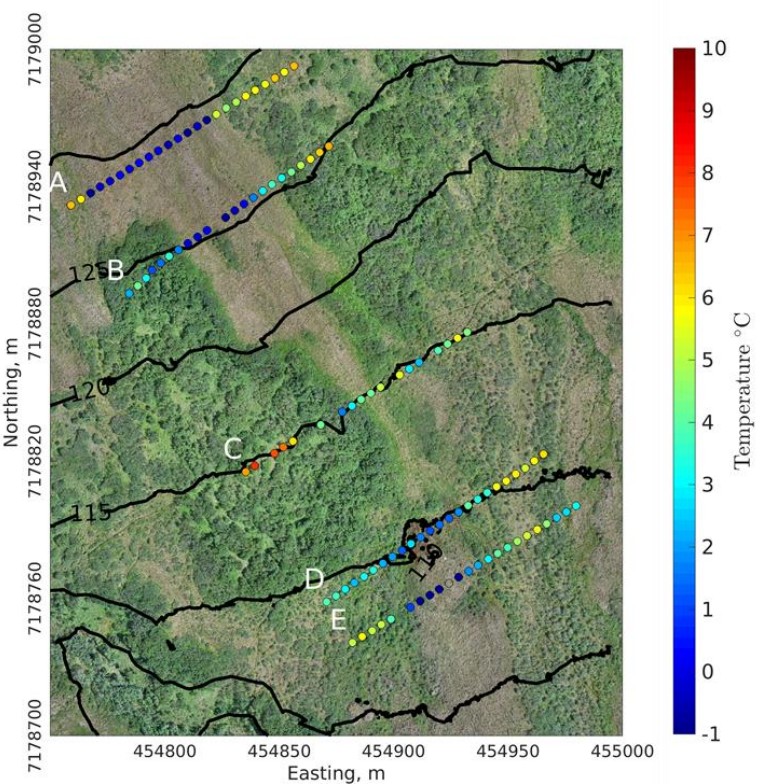

**Figure 2: Temperatures at 0.8 m depth extracted from the DTP dataset collected on July 17 and topographic isolines for every 5 m of elevation (in meter amsl.).**

Figure 3 shows each transect in a relative coordinate system, in order to accommodate visualizing topography, vegetation type, soil moisture, and the July 17 vertically resolved DTP data together. Based on vertically resolved DTP data, soil temperature close to or below 0°C at 0.8 m depth and with a trend in temperature with depth going clearly toward negative temperature values indicates the presence of near-surface permafrost. This is the case between 15 and 75 m along transect A, 40 and 75 m along transect B, and 35 and 65 m along transect E (black rectangles in Figure 3). The shallowest thaw layer (0.45 m) is observed at 45 m along transect E. This location also corresponds to the lowest temperature at 0.8 m depth, once we extrapolate soil temperature to this depth. The vertical thermal gradient in the top 0.8 m of soil, and the measurements of water content in the top 0.3 m using a TDR, show sharp changes at a similar location, while their values are not always positively correlated (Figure 3). Locations identified as near-surface permafrost along transects A and B show high soil

water content in the top 0.3 m, while a location identified as near-surface permafrost along transect E shows low water content in the top 0.3 m.

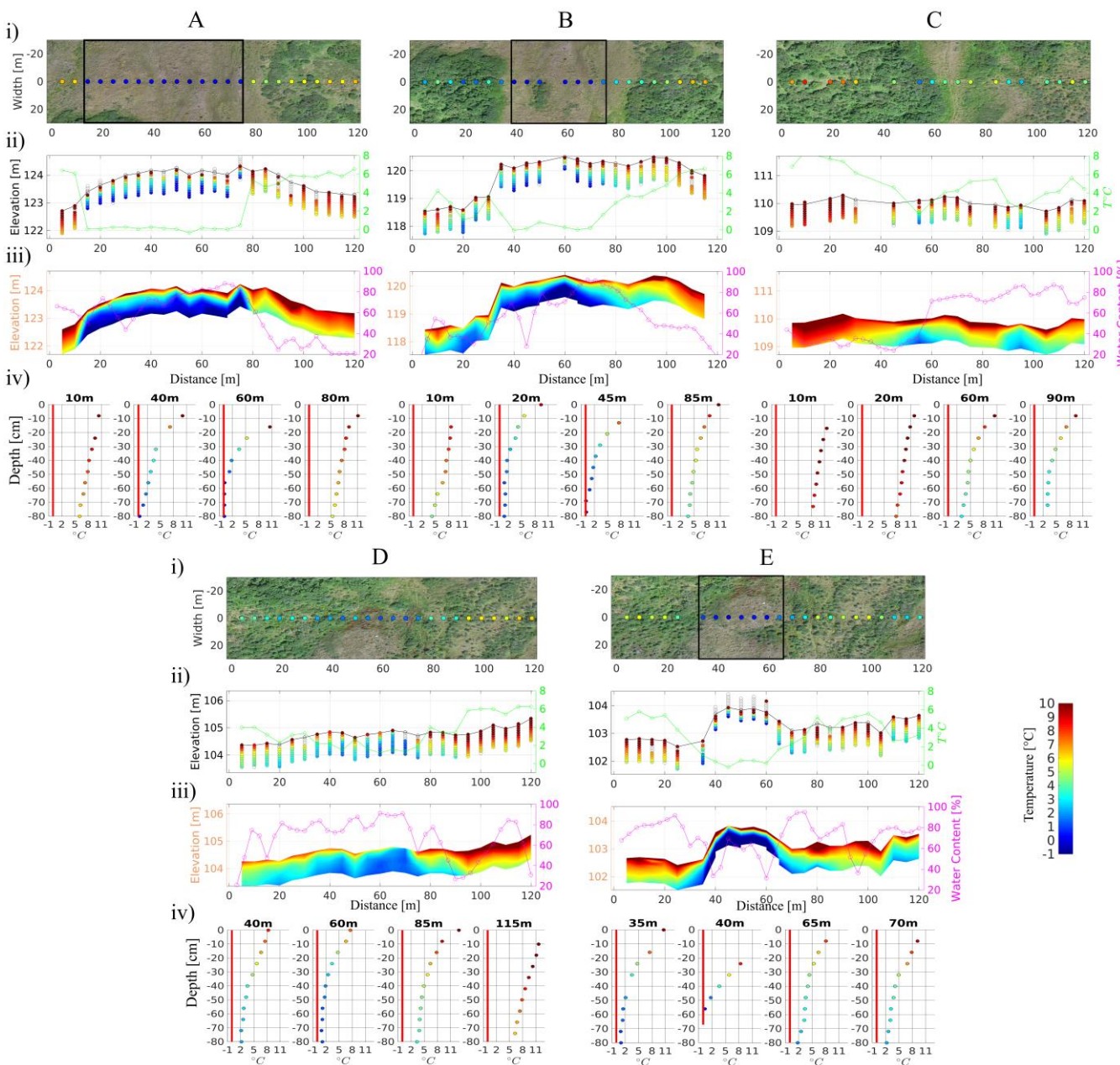

Figure 3: DTP data collected on July 17, 2017, along A-E transects. (i) Aerial view of the A-E transects in relative coordinate systems, overlain by temperature at 0.8 m depth and consequently identified near-surface permafrost areas (black rectangles), (ii) DTP temperature profiles and DTP temperature at 0.8 m depth (green line), (iii) Interpolated temperature map of the first 0.8 meter with topography and surface water content, and (iv) temperature profiles at selected locations along the transects.

These abrupt lateral changes in soil temperature also correspond to changes in vegetation and topography (Figure 2 and 3). Figures 2 and 3 suggest that in general the soil temperatures at 0.8 m depth are the highest under tall-shrub-covered areas (up to 7.5°C), and the lowest under graminoids and dwarf-shrub-dominated areas (down to 0.2°C or below). This general trend is modulated by or intertwined with many factors. The topographic lows along each transect tend to correspond to higher soil temperatures at 0.8 m depths than the topographic highs (Figure 2). These topographic lows correspond here to preferential drainage paths crossing the transects perpendicularly and are possibly related to ground erosion and/or ground settlement, as well as to locations with higher accumulation of snow during the winter.

In Figure 4, the soil temperature data are compared to the two long-term thermal monitoring stations (Stations 9 and 2 in Figure 1) to evaluate any potential limitations in interpreting the one-time DTP dataset, and to evaluate the value of acquiring spatially dense DTP data. Note that while the DTP system can be deployed for monitoring purposes, here we concentrated on first evaluating its value by acquiring an initial large dataset in July 2017 and later repeating measurements along transects B and C in September 2017. Figures 4a and 4b show the soil vertical profile of temperature from the long-term monitoring stations measured every mid-month (i.e., every 15th day of the month) from January to September 2017. Figure 4c shows the soil vertical profile of temperature from the long-term monitoring stations overlain on the DTP system measurements along transect A to E in July and transect B and C in September. Soil temperatures at 0.25 m depth or deeper at Station 9 are lower than at Station 2 all year around. While no temperature measurements are located deeper than the permafrost table at Station 9 and 2, the trend in the temperature data indicates that the permafrost table is likely between 1 and 2 m at Station 9 and much deeper or absent at Station 2. Soil temperature data at Station 2 and 9 are in the upper and lower range of temperature values observed using the DTP system, respectively. Several locations in the DTP dataset show higher and lower temperatures and represent end members in the system.

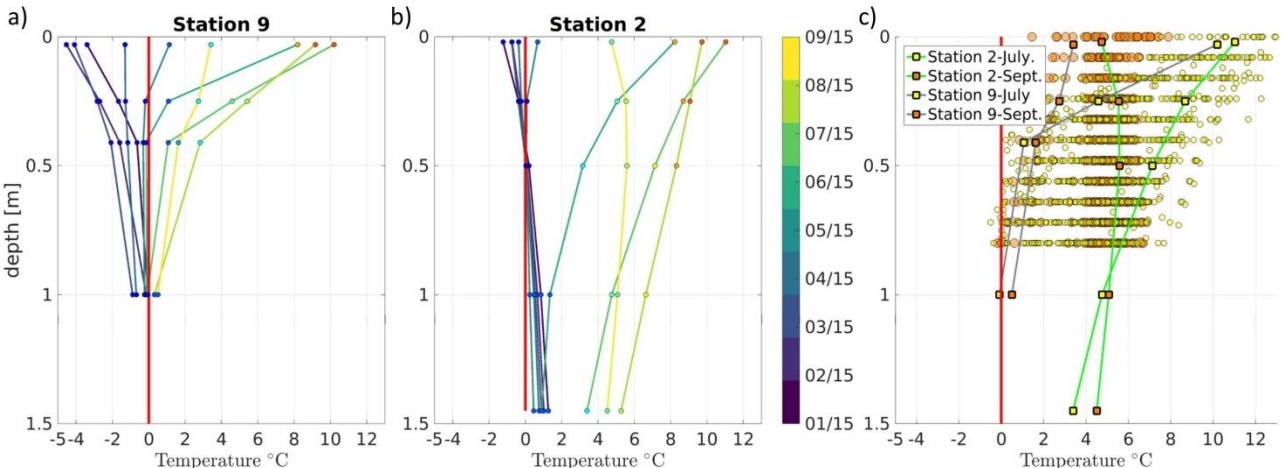

**Figure 4: Temperature profile at Station (a) 9 and (b) 2 at every mid-month from January 15, 2017 to September 15, 2017). The zero Celsius degree line is in bright red. (c) Comparison of temperature measurements at Stations 2 and 9 (solid lines) with DTP measurements (circles) along transects A to E on July 17 (yellow) and along transects B and C on September 20 (orange).**

The surficial seasonally frozen layer at Station 2 that developed during the freezing season has entirely thawed before June 15. This observation confirms that the DTP dataset acquired on July 17 is not threatened by potential misinterpretation of near-surface permafrost where a seasonally frozen layer over a thick (> 1 m) perennially unfrozen soil is present. At locations where a perennially unfrozen layer is thin or absent, the DTP temperature measurements acquired on July 17

indicate trends in permafrost table depth across the landscape although a precise estimate of the permafrost table depth cannot be obtained at this time of the year. Further, Stations 2 and 9 both show that the vertical profile of temperature has a sharp gradient in the top 0.25 to 0.5 m depth, while at greater depth the temperature has an increasingly asymptotic behavior. This expected behavior underlines the importance of measuring temperature with highest vertical resolution close to the surface, while still acquiring measurements deeper than 0.5 m where asymptotic trends in soil temperature with depth are

more present and strongly informative on the deeper thermal regime.

In Figure 5, the DTP dataset collected along transects B and C on September 20 is displayed in a similar way to the DTP dataset collected on July 17 in Figure 3. Figure 5iv shows four of the DTP vertical profiles of soil temperature acquired on September 20 and the corresponding ones collected on July 17. The DTP soil temperature at 0.8 m depth measured on September 20 shows a very similar spatial trend (although with different absolute values) to that measured on July 17, with

clearly identifiable near-surface permafrost locations (Figure 5i and Figure 3i). The air temperature in September is lower than in July, and thus the top 15 cm of soil or more (depending on the location) are colder on September 20 than on July 17. Compared to July 17 DTP data, the DTP vertical profiles on September 20 show lower thermal gradients, as expected at the end of the summer season. The majority of locations show higher soil temperature in the 0.5 to 0.8 meter depth interval on September 20 than on July 17, while other locations already show the effect of the decrease in downward heat flux at the end

of the summer in this interval. The spatiotemporal difference in DTP vertical profiles between September 20 and July 17, and between the various locations, underlines the complexity of how the heat flux is mediated by soil thermal characteristics in the investigated depth interval, as well as by surface and vegetation properties and the thermal regime at deeper depth than 0.8 m.

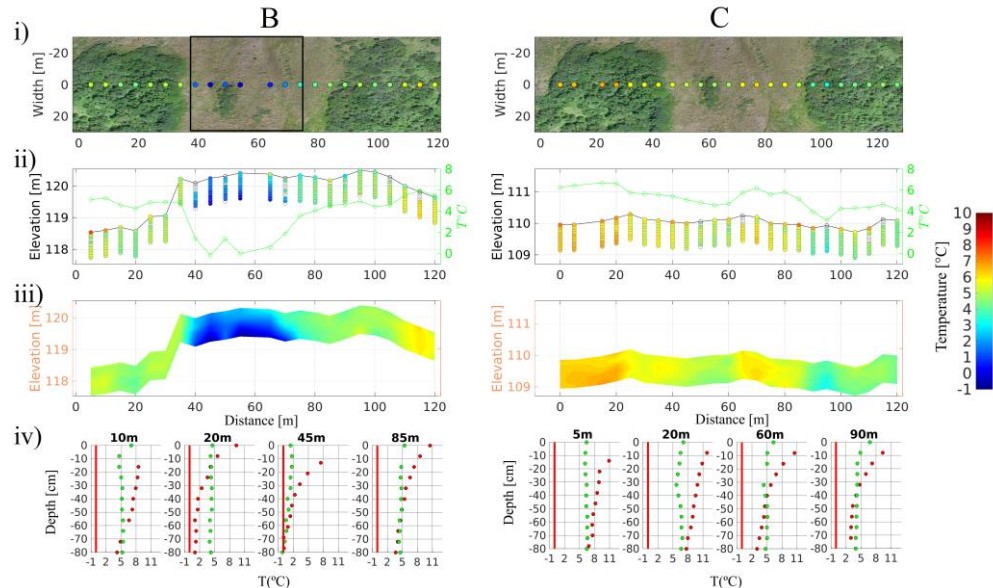

**Figure 5: DTP data collected on September 20, 2017 along B and C transects. (i) Aerial view and identified near-surface permafrost area (from Figure 3) overlain by soil temperature at 0.8 m depth (green line), (ii) DTP temperature profiles and DTP temperature at 0.8 m depth, (iii) Interpolated temperature map of the first 0.8 meter with topography, and (iv) temperature**
**profiles (green dots) at selected locations along the transects and compared to collocated temperature profiles acquired on July 17 (red dots, from Figure 3).**

Figure 6i allows comparison of DTP data at 0.8 m depth with the ERT data acquired along the same transects at the same time on July 17, 2017. This comparison enables us to assess both the value and the limitations of the DTP system, and in particular to interpret the vertical extent of near-surface permafrost, based on shallow temperature measurements. The
ERT transects indicate the presence of large and shallow resistive bodies (up to $10^4$ Ohm.m, with highest resistivity values in the top 5 m) along transects A and B, which are quite isolated and have sharp lateral resistivity variations. Less resistive zones (approximately lower than 500 Ohm.m) surround these resistive bodies. The most conductive areas (around 300 Ohm.m) are located close to the surface and in some cases above the resistive bodies. Transects A and B exhibit the same type of resistivity distribution. Transects D and E have greater similarity to each other than to A and B, except for the
shallow resistive area in the middle of transect E. Transect C has a conductive area positioned between deep resistive zones. The near-surface permafrost regions identified in the DTP data (black rectangles in Figure 3) are collocated with the presence of shallow resistive bodies in the ERT data. This is the case between 15 and 75 m along transect A, 40 and 75 m along transect B, and 35 and 65 m along transect E. While thermo-petrophysical analysis of the ERT data is well beyond the scope of this study, here both the presence of resistive bodies in the ERT data and the collocated presence of low temperature
values observed in the top 0.8 m suggest the presence of near-surface permafrost locations.

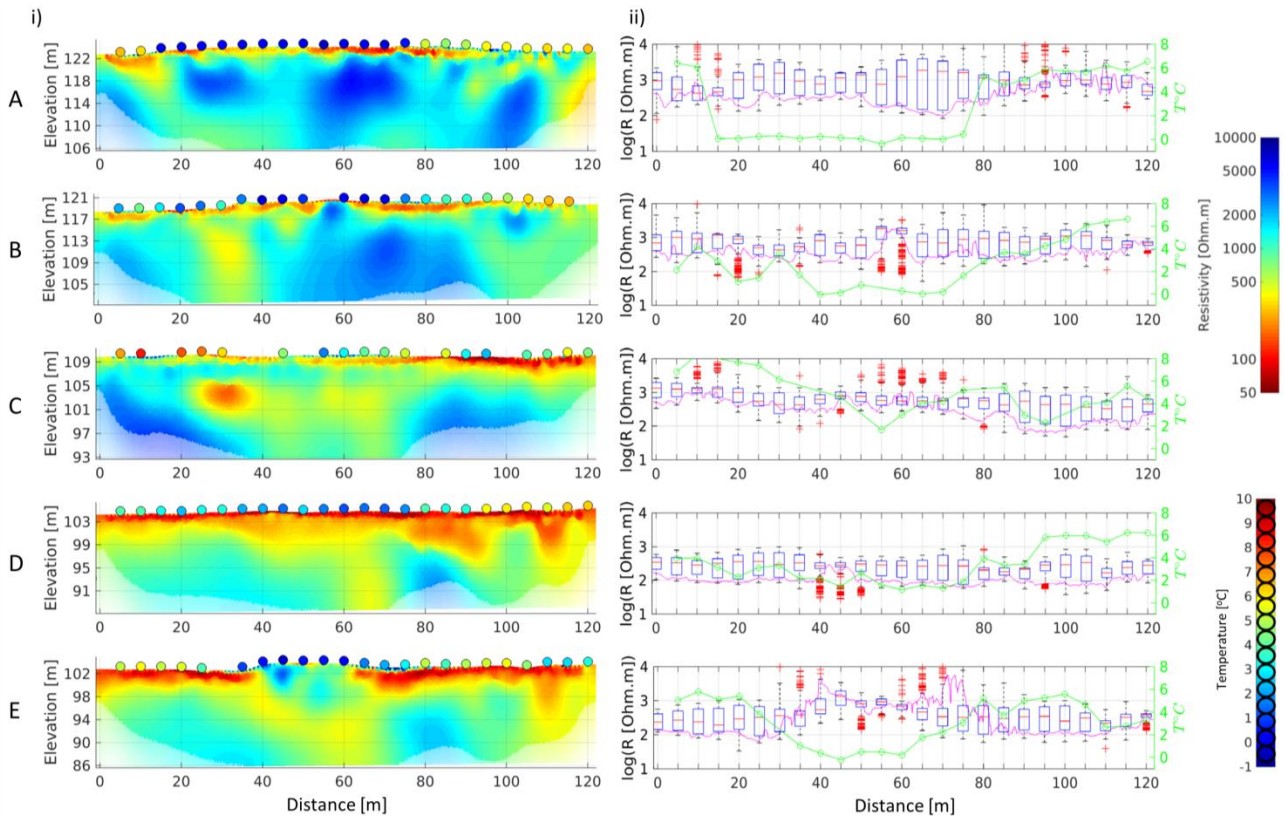

**Figure 6: (i) ERT data with soil temperature at 0.8 m depth (extracted from the DTP dataset from July 17, 2017) shown at the top of each transect, and (ii) boxplots of the resistivity values in the top 7 m depth, vertically-averaged resistivity in the top 0.8 m (purple line) and temperature at 0.8 m depth (green line).**

The DTP provides the temperature gradient in the top 0.8 m of soil, indicating the potential presence of permafrost at or deeper than this depth interval, while an increase in soil resistivity in the ERT is located where the ground is mostly frozen. Thus, both approaches provide different but complementary information about the depth of permafrost. Figure 6ii shows the DTP dataset at 0.8 m depth, the top 0.8 m depth average resistivity values for each DTP location, and the range of resistivity values (displayed in a boxplot format) observed over the top 7 m depth in the ERT at each location. The top 0.8 m depth average resistivity value shows a spatial variability relatively similar to the soil-water-content data, with the exception of between 0 and 45 m along transect C. At near-surface permafrost locations, the temperature profiles down to 0.8 m depth indicate temperature decreasing with depth toward the freezing point (generally located deeper than 0.8 m depth), while the ERT at similar depth is widely sensitive to the water content in the thawed layer. The lateral variability in the DTP data at 0.8 m depth is more consistent with the ERT data when considering the full range of resistivity values observed over the top 7 m in the ERT. Finally, we note that although lateral changes are observed at relatively similar locations in the various properties presented in this study (Figure 3, 5 and 6), the correlation coefficient for each possible combination of properties across the site is always smaller than 0.5, indicating that no clear linear relationship exists.

## 5 Discussion

In this section, we discuss the information contained in the DTP data and its potential, when coupled with other ground- and aerial-based geophysical datasets, for evaluating the distribution of permafrost.

### 5.1 Spatial Distribution of Near-Surface Permafrost

In environments where topography, soil water content, vegetation, snow thickness, soil-organic-matter content, and other parameters vary strongly over meters to tens of meters, understanding how these properties individually or in combination modulate the heat and water fluxes that influence permafrost distribution and temperature is very challenging. A key advantage of the DTP system is its ability to directly collect spatially dense (horizontal and vertical) soil temperature data. The DTP dataset discussed here provides information important to understanding the ecosystem functioning at the

investigated site. First, it directly provides clear identification of near-surface permafrost locations, such as between 15 and 75 m along transect A, 40 and 75 m along transect B, and 35 and 65 m along transect E (Figure 3). The comparison between the main survey done on July 17 (Figure 3) and a second survey limited to transect B and C on September 20 (Figure 5) confirms the general spatial trend in soil temperature and identification of near-surface permafrost locations. The comparison also shows that the lateral extent of the permafrost body along transect B is a few meters smaller than initially identified on

July 17. It is clear that the identification of near-surface permafrost is less prone to uncertainty when performed at the end of the summer. Here, the observed variations along the sides of the near-surface permafrost also indicate the particularly complex spatiotemporal variability in lateral and vertical thermal fluxes occurring at these locations.

     The DTP system also provides high-enough spatial resolution to observe possible relationships between soil temperature and topographic features, soil water content, and vegetation type. We find that in the study area near-surface permafrost

bodies are always located under topographic highs (at various scales), as seen in Figure 3. This co-variability is in agreement with the expectation that ground settlement will be limited in the presence of near-surface permafrost compared to surrounding locations with deeper or no permafrost. It is also consistent with the fact that topographic lows formed through ground settlement or erosion tend to have thicker snow cover during the winter, which provides more soil insulation and leads to warmer soil thermal conditions compared to topographically high regions (e.g., Wainwright et al., 2016).

Also, the presence of near-surface permafrost identified from the DTP dataset is strongly correlated with the presence of graminoids and/or lichens and dwarf-shrub-covered areas. The graminoid-dominated area crossing transect A (between 15 and 80 m) and B (between 40 and 80 m) can now be considered as encompassing a near-surface permafrost body (Figure 2). The lichen and dwarf-shrub region located along transect E (between 25 and 65 m) is also clearly identified and shows the lateral extent of the near-surface permafrost there.

Further, the soil moisture data suggests that the thaw layer above the near-surface permafrost bodies identified in transects A and B (and collocated with the presence of graminoids) is very wet if not fully saturated (Figure 3). We interpret this high-soil-water content in the thaw layer as associated with the limited drainage capacity imposed by the topography and

the presence of near-surface permafrost. Given the thermal properties of water, one could expect to observe thicker soil thaw layer where soil is fully water saturated compared to dry soil (Tran et al., 2017). This relationship is clear when comparing the very shallow thaw layer above permafrost (~0.45 m depth) observed at 45 m along transect E to the area with deeper thaw layer above near-surface permafrost along transects A and B. Indeed, the driest soil is observed at 45 m along transect

E (Figure 3). While this relation is observed between these two locations where near-surface permafrost is present, the soil water content in the thaw layer above the near-surface permafrost along transects A and B is higher than at many other locations where near-surface permafrost is absent. The influence of soil water content on thermal parameters and regime is certain but complex, because of water phase changes and soil water content temporal variability controlled by hydraulic parameter and thermal hydrology.

DTP measurements and related identification of near-surface permafrost locations is consistent with ERT data, where high resistivity bodies are identified at similar locations along the transects and at shallow depth (Figure 6). The high resistivity values observed in the shallow depth (top 4 m) in the ERT are generally located deeper than where the 0°C is expected from the DTP data. This is not unexpected: it can result from (i) smoothness in the ERT inversion, (ii) the presence of still large unfrozen water content at temperatures between -2 and 0°C, and/or (iii) particularly large unfrozen water

content during freezing where the total water content is high.

The combination of DTP data and ERT is also valuable for investigating the subsurface vertical heterogeneity in regions where near-surface permafrost bodies are present. The identified near-surface permafrost bodies extend vertically to about 15 m depth along transects A and B, but much less along transect E, based on the ERT data (Figure 6). Assuming constant salinity and homogeneous lithology, the ERT suggests that the coldest and most strongly frozen regions of near-surface

permafrost bodies occur at depths of about 6 m along transects A and B, and at about 2 m depth along transect E. The DTP data and thaw layer thickness indicate shallowest top of permafrost along transect E, which is consistent with the ERT data. While temperature in the DTP data could be valuable by itself, adding the ERT in this case enables us to clearly identify locations where near-surface permafrost is thin.

## 5.2 Beyond the Identification of Near-Surface Permafrost Locations

While the value and promise of the DTP system for improving our understanding of near-surface permafrost distribution and related shallow lateral and vertical variation in temperature regimes have been demonstrated in the previous section, the variations in soil temperature, where deep permafrost—with a perennially thawed zone above it—or no permafrost is present, are more difficult to interpret.

When surface water content is high and the permafrost table is deep or permafrost is absent, the thermal profile shows

relatively low temperatures right below the surface, with a relatively small vertical temperature gradient. This is the case between 0 and 70 m along transect D and between 0 and 30 and 65 and 120 m along transect E (Figure 3). These locations are at the flat toe of the hillslope, with the groundwater table at or very close to the surface. The soil temperature at these locations is around 1-2°C at 0.80 m depth, and possibly decreases below 0°C at much greater depths. We interpret the large

soil water content and some water inundation to be maintaining a relatively homogeneous and stable shallow temperature regime, owing to the water's high heat capacity. In such a wet environment, the latent heat effect is likely reinforced as well. The melting pore ice can absorb additional heat and slow the rate of temperature increase. Similarly, as temperature decreases in autumn, freezing pore water releases latent heat and slows temperature decline (Osterkamp and Romanovsky,

1997; Shojae Ghias et al., 2017). At such locations, the DTP data indicate that a perennially thawed zone may be present, along with the possible presence of deeper permafrost.

Another interesting case is along transect C (Figure 3) between 0 and 30 m. At this location, the DTP data show a very small thermal gradient between the surface and 0.8 m depth, where the temperature is still as high as ~7°C. In addition, this location has a ground elevation that is slightly higher than the rest of the transect, is relatively dry at the surface, is situated

in tall shrub area where snow depth may be large during the winter, and is interpreted to have some rocky soil based on probe installation. This relatively dry and rocky location suggests that the soil in this region has a low heat capacity, which would decrease the ability of soil to maintain its temperature in the surveyed depth interval and produce a small thermal gradient as observed here. This is confirmed by the September 20 DTP data between 0 and 30 m along transect C (Figure 5iv) showing lower soil temperatures in the entire 0-0.8 m surveyed depth interval (compared to July 17), which indicates

that the lower air temperature in September likely had already a strong influence on the entire temperature profile. This is different from most of the other locations, which show an increase in soil temperature at 0.8 m depth between July 17 and September 20. In addition to dry and rocky material that could explain the limited ability of soil to maintain its temperature in the surveyed depth interval, a change in soil thermal parameters below the surveyed depth interval may further impede the transfer of heat to a deeper layer. At this time, we cannot confirm if the deeper layer, which indeed is resistive in the ERT

data, were permafrost or bedrock. The end of transects A and B most likely exhibit the same physical process discussed above, in which shallow temperature is relatively high and water content relatively low, producing higher uncertainty regarding permafrost presence at depths greater than 4 m.

The aforementioned results, and the complexity involved in how heat flux is modulated by various surface properties and soil heterogeneity, underscore the importance of surveying numerous locations, and measuring soil temperature profiles

down to 0.8 m depth or deeper, ideally over time. The DTP system and the long-term monitoring stations 2 and 9 show that measuring only the very-shallow surface temperature (first 30 cm) is insufficient to evaluate deeper thermal regimes (Figure 4). Temporal variations in soil temperature in the top 10 to top 30 cm can be significant over short periods of time (up to hour scale), and not representative of deeper thermal regimes, especially in dry areas. In addition, the strong decay in temperature in the top 30 cm can produce larger error within temperature data, in case of uncertainty as to the exact depth

where the point sensor is located. This expected behavior, based on the physics of thermal flux, confirms the need for measuring temperature with high vertical resolution close to the surface as well as at depths deeper than 0.5 m, where asymptotic trends in soil temperature at depth are more present. In addition, the comparison of July 17 and September 20 surveys (Figure 5) shows that observing the system over time provides, as expected, very valuable information to go beyond the identification of near-surface permafrost. The DTP dataset will clearly benefit from year-round acquisition of soil

temperature measurements at different depths, which are key for the interpretation of both surface energy balance and deeper thermal characteristics and regimes. High vertical resolution close to the surface (Figure 3) has the potential to provide information on freezing or thawing fronts in shoulder seasons, as well as on the influence of organic layers on deeper thermal regimes.

Finally, based on the long-term monitoring station data (Figure 4), we expect that obtaining the fluctuation in temperature over time using the DTS will improve the ability to evaluate various regimes and factors. We also expect the time-lapse soil measurements to be useful as input to inverse modeling techniques focused on estimating soil thermal parameters and possibly soil-organic-matter content (e.g., Nicolsky et al., 2009; Tran et al., 2017).

## 6 Conclusion

This study describes a novel strategy referred to as Distributed Temperature Profiling (DTP) to quantify the near-surface soil-thermal state at an unprecedented number of depths and locations and test this approach for the delineation of near-surface permafrost regions in a discontinuous permafrost environment by moving the DTP system sequentially across the landscape. To our knowledge, this is the first time that a thermal characterization approach has provided such high vertical and lateral density in measurements. The low cost, portability, and ease of deploying the DTP system makes this method

efficient for investigating permafrost spatial heterogeneity, particularly where significant lateral variations in soil temperature occur over meters to tens of meters. As shown, the densely spaced DTP data compared well with classical thermal measurements and provided much higher spatial resolution. Possible deployment of DTP systems in a time-lapse mode will also enable high temporal resolution and larger temporal coverage.

In this study, the DTP dataset has shown to be particularly valuable in delineating the presence of near-surface

permafrost. When combined with other approaches, the DTP was also useful for evaluating correspondences between summer soil temperature, plant type, topography, soil water content, and deeper subsurface structure. While decoupling the control of these and other factors (e.g., organic content and snow depth) on thaw layer thickness and the depth of permafrost table is complex and beyond the scope of this paper, our data indicate that changes in soil temperatures often correspond to changes in topography, vegetation, and soil moisture. Near-surface permafrost identified in the study area using the DTP

data is primarily collocated under topographic highs and under areas covered with graminoids or lichens and dwarf shrubs. The results provide new insights about the presence or absence of near-surface permafrost bodies.

The simple and low-cost DTP strategy holds promise for improving our ability to quantify local permafrost distribution and to explore interactions between complex Arctic ecosystem properties and processes. In particular, coupling various observations has the potential to advance strategies to estimate permafrost distribution from remotely sensed information,

including topography, vegetation, and possibly surficial moisture characteristics. This study opens the door to such quantification, although many challenges remain. For example, while co-variability is observed between near-surface permafrost and topographic highs and absence of tall shrubs, multiple factors need to be accounted and estimated for

possibly identifying these from remote sensing data. It includes automated extraction of topographic information at various scales, as well as the identification of physical vegetation, snow and soil characteristics.

Ongoing developments in the DTP system include the miniaturization and cost-reduction of the data-logger, improvement and cost-reduction of the temperature probe fabrication, ability to produce probes with variable length and vertical resolution, development of open-source software and hardware to encourage community-based development and deployment, and automated acquisition and data transfer for long-term monitoring purposes. These developments will enable the deployment of low-cost DTP to measure and record ground temperatures all year-round and with a number of probes that are well beyond that described in this study. The potential of this method for characterizing and monitoring a variety of near-surface heat and related water dynamic processes is significant for informing investigations aimed at quantifying water infiltration, evaporation, biogeochemical processes, hyporheic exchange, snow-melt dynamics, and permafrost evolution.

### Data availability

All data used in the analysis presented here are available by contacting the corresponding author or at https://doi.org/10.5440/1559886 (B. Dafflon, E. Leger and S. Hubbard. 2019. Characterization of Soil Thermal and Electrical Properties along Multiple Hillslope Transects at Teller Road Site, Seward Peninsula, Alaska, 2017. Next Generation Ecosystem Experiments Arctic Data Collection, Oak Ridge National Laboratory, U.S. Department of Energy, Oak Ridge, Tennessee, USA).

### Author contributions

EL and BD developed the acquisition strategy, conducted the tool development and the data acquisition and analysis, and prepared the paper. YR assisted with the development, acquisition and analysis. CU, JP and SB assisted with data acquisition. VR and SH assisted with developing acquisition strategy and preparing the paper.

### Competing interests

The authors declare that they have no conflict of interest.

### Acknowledgments

We acknowledge the assistance of Todd Wood, Paul Cook, and Alejandro Morales of the Geoscience Measurement Facility (GMF, gmf.lbl.gov) at LBNL for building the DTP prototype system. The probe sleeve and filling material were partly influenced by the work of Bill Cable (UAF, Alaska).

**Financial support**

The Next-Generation Ecosystem Experiments (NGEE-Arctic) project is supported by the Office of Biological and Environmental Research in the DOE Office of Science. This NGEE-Arctic research is supported through contract number DE-AC02-05CH11231 to Lawrence Berkeley National Laboratory.

**Review statement**

This paper was edited by Ketil Isaksen and reviewed by two anonymous reviewers.

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
