# Peer review of "A distributed temperature profiling method for assessing spatial variability of ground temperatures in a discontinuous permafrost region of Alaska"

_The Cryosphere, 2018_

## Referee Comment (RC1) · Anonymous Referee #1 · 17 Mar 2019

Dear authors,

I have read the manuscript submitted to The Cryosphere and think that it provides an interesting approach to surveying shallow ground temperatures, allowing to obtain a good spatial snapshot, which improves the understanding of detailed variability. The manuscript is well-written, although sometimes needing more objective comments and some polishing. There is a strong focus on the low cost of the instrumentation, but that is really not very novel, since this type of devices have been developed and applied by numerous teams over the last two decades. However, they are becoming increasingly

cheaper due to the reduction of hardware costs, development and miniaturization of sensors and datalogger and higher availability of open-source software. I therefore suggest that the authors put more focus on the novelty of the concept of moving the probes along the measuring site, rather than on the monitoring system itself, especially since this approach (monitoring) has not been conducted in this case-study.

Still on the measuring method (profiling), this type of approach was frequently used on local and urban climate studies, at least until the 1990's, for measuring air temperatures across large areas, with sampling measures at specific sites, which were then temperature-corrected for time, in order to allow for comparison. Such a correction might even be interesting to be done here, to make use of the near-surface temperatures, which are affected by the diurnal changes (>25 °C following your manuscript). Examples of such application should be mentioned, since they link to the proposed method, that can be seen as precursor of the method. Here are some examples, but others also exist: https://doi.org/10.4113/jom.2010.1112, https://doi.org/10.1007/s00704-005-0152-1.

Temperature profiling has also been done by other authors on permafrost settings. For example, Goyanes et al (2014) (http://dx.doi.org/10.1016/j.geomorph.2014.04.010), measured temperatures from 5 to 70 cm depth in a grid in Deception Island to detect the effect of geothermal anomalies on permafrost distribution. They have also compared the results with ERT surveying. Although not with dataloggers and with a scarcer number of temperature measuring depths, but the approach is comparable to the one presented here, especially since in this manuscript the datalogging has not really been used. So, please check references and extend the review also to CALM-related publications, since I would think this has been done elsewhere.

As you will find in the detailed review below, I think that you need to improve the characterization of the site in order to better assess the results. I think a larger-scale map derived from the UAV survey (or high resolution satellite) would be helpful, and also a window showing the setting of the studied slope in the watershed context. A discussion on the spatial variability of soil characteristics is needed, especially since the site is not homogeneous (as you mention, there are possible rock outcrops). Following this rationale, a description of the geomorphic units is also lacking, since concavities and convexities, may possibly be explained by different dynamics and also reflect soil (deposit?) types and hence will have an effect on soil temperature.

I also think you should be more cautious in what concerns to permafrost distribution, since it seems that you present no single direct observation of permafrost, other than the indirect measures by ERT. Please clarify this and indicate if there are other observations that show the characteristics of permafrost at the site or in its vicinity.

As a conclusion, I think the manuscript is of good quality and should be published after a thorough review. The results show that the approach can be of wider application, especially with multiple datalogging systems installed, but the main-added value related to the way you apply the surveying.

Detailed comments: Title: I think the title is probably too ambitious for the contents. I would suggest a more focused title, such as "A distributed temperature profiling method for assessing spatial variability of ground temperatures (Nome, Alaska)". Page 1, l. 10. Please clarify/rephrase the sentence, since temperature does not contain information on the properties modulating the soil thermal flux; it rather reflects these properties. Page 1, l. 18. Replace AK by Alaska. The acronym is poorly known outside the US. Page 3, l. 25. I would suggest deleting the mention to Arctic permafrost regimes and write only "permafrost regimes", and add a few citations from non-Arctic regions (e.g. mountain and Antarctic). Page 4, l. 6. I reccomend extending this review to non-Arctic permafrost sites, since significant advances have been presented, for example, in the European Alps. Page 4, l. 19-20. Consider rephrasing the sentence, since it doesn't make too much sense to distinguish between soil minerals themselves and the organic component that may be mixed with the soil minerals, since you are aiming here to monitor the soil as a whole. Page 4, l. 35 – delete "low-cost". It has been mentioned earlier. Page 5, l. 7 – Replace AK by Alaska. Page 5, line 8 – Consider replacing "This

"Teller watershed" by "The studied watershed along Teller Road" Page 5, line 16-17 – Delete cumulative twice; it is not necessary, since annual precipitation data is always cumulative. Page 5, lines 20-23 – You should rephrase and clarify if the most frequent situation is a thaw depth greater than 1 m, or a thaw depth of 0.4-0.6m. It is not clear (at least for the reader at this stage of the manuscript) why you are emphasising on such thaw depths, when there are also deeper ones. Page 7, l. 1-5 Simplify text and make it more objective. Excessive use of adjectives associated to costs, etc. Page 7, line 17 . Where were those probes installed and what are their characteristics? Page 7, line 15-20. I find it difficult to accept that a system that was specifically designed to be tested for measuring soil temperatures, especially using such minute and fragile sensors, has not been calibrated. It is also not clear why temperatures at 80 cm depth were used, since not all sites were measured down to that depth. How many sites were extrapolated and what is the quality of the extrapolation? The assumptions that "no sensor-specific calibration curve could easily be defined while ensuring that it would increase sensor accuracy significantly" needs a better framing. If the system is to be applied in the future, calibration is needed and mentioning that it could not have been done because it is 80 cm long, reveals some limitations for future developments. Page 8, l. 1 – Please clarify the setup. Is the pvc tube buried vertically with the sensors at different depths? What is the thickness of the tube? Page 8, line 26 – indicate where in the Arctic, since conditions are variable. Page 8, l. 31 – replace "topographic trends" by "topography". Page 9, l. 1 – replace "inferred" by "prepared". Page 9, line 3. Delete "More details on UAV-based imaging..." and keep only the reference to Dafflon et al 2016. Page 9, l.9-11 – delete. Not necessary. Page 9, l.-13-15 – This should be moved to the methods. It answers the question raised above. Page 11 l. 6 – You mention vegetation and topography, but for such a detailed spatial and vertical analysis, you will certainly need to characterize geomorphology and soil type to better understand spatial variability. Add these characteristics to the study site description. Page 12, l. 2-5 – consider revising: "quite perpendicularly to the general slope aspect...". Improve phrasing "...possibly related to ground erosion and/or ground settlement". Why? This

should be better discussed under the presentation of the study site. Page 12, Line 12-13 – "...at many other locations for each depth..." - Clarify. Page 12, line 14 – "with a few exceptions for the shallowest measurement..." – clarify. Page 12, line 24-25 – Clarify what you mean with this sentence. Page 14, line 16 – How can you associate directly the presence of permafrost with temperatures at of below 0.2 °C in mid-july? Please rephrase the sentence carefully, since you are using ERT data as a proxy for permafrost, which should work fine, but check the phrasing. Page 14, line 20 – I would prefer indicating that the temperature values "suggest" the presence of permafrost, rather than "indicate", since you have no direct observations of permafrost along the transects. Page 15, lines 14-15. Why does this happen? A scatterplot or at least R2 values could help understand the described co-variability, since the figure is really small. Page 18, line 22 – Replace "temperature flux" by "heat flux". Page 19, line 9 – Please rephrase the sentence reconsidering the novelty. Page 19, line 32 – You mention remote sensing data, but I think you mean satellite data, since with UAV surveys one can very well characterize structure of shrubs (multispectral, LIDAR and low elevation flights at multiple angle surveys).

Figure 1 – Consider beginning the caption with a title that applies to both windows (i.e. Location and general setting of the study area) and only after mention "a) Aerial view...". Enlarge the dots of the two stations which are very difficult to depict in the image. Add graphic scale in Figure A. Although the UTM coordinates are there, they may not be self-explainable to many readers. A larger map of the UAV survey with topography would be important ion order to assess the topographical and geomorphological setting of the slope section under analysis. That is very important to frame possible subsurface flow paths, geomorphic units and processes that may be active and which influence soil properties. Figure 2. Add graphic scale. In caption, remove the mention to being overlain in figure 1. Figure 3 – The permafrost areas identified with the rectangle are based in which data? The date is still not the one of maximum thaw depth, I suppose. Please clarify in the caption and in the text. The Y-axis for TDR is difficult to read. Check colour and font size. You mention that temperature above 25 cm were influenced by

diurnal oscillations. Hence, I think they should not be plotted. Figures 4 and 5 – If only the temperatures deeper than 25 cm should be used for comparison, I would suggest removing them from the figures.

---

## Referee Comment (RC2) · Anonymous Referee #2 · 21 Jun 2019

The paper entitled 'Distributed Temperature Profiling System Provides Spatially Dense Measurements and Insights about Permafrost Distribution in an Arctic Watershed' by Léger et al. provides a highly interesting strategy for obtaining spatially and temporally dense soil temperature measurements at flexible spatial scales. The strategy is tested and evaluated in an Arctic watershed near Nome in Alaska. This is a well documented and thorough study that contains valuable results. The methodology is sound and the assumptions are clearly identified. It is a good paper, which also brought some new data and knowledge on permafrost behavior in Alaska. Publication of this kind of paper

will be very timely and beneficial for researchers working in the same field, as well as for many other researchers conducting a wide spectrum of environmental studies. The paper needs only minor revision, mainly regarding site description and readability of the figures.

P2, L17 include "soil" » characterizing and monitoring soil temperature predominantly...

P2, L21 UTL-3 Scientific Datalogger (https://www.geotest.ch/en/expertise/measuring-monitoring/utl-3-temperature-datalogger.html) is frequently used in Europe and could be included in the list of loggers

P2, L18-26 Consider to include one sentences briefly describing analog and digital temperature acquisition systems for ground temperature monitoring

P5, L13-14 and in presentation of results/discussion: I miss specific details related to soil type(s) at the study site. This is important in relation to the spatial variability of soil temperatures and corresponding interpretations. Some kind of information, e.g. soil profiles/variability in soil characteristics at the DTP sites could also be useful.

P9, L15 Suggest to replace "coldest temperatures" with "lowest temperatures". In also other parts of the manuscript (e.g P12, L2; P12, L13) "cold/warm temperatures" are used. Better write "low/high temperatures".

Figures: The readability and quality of the figures should be improved. Please read carefully the guidelines regarding figure composition, figure content and figure captions provided by TC at https://www.the-cryosphere.net/for_authors/manuscript_preparation.html

---

## Author Comment (AC1) · 25 Jul 2019

"I have read the manuscript submitted to The Cryosphere and think that it provides an interesting approach to surveying shallow ground temperatures, allowing to obtain a good spatial snapshot, which improves the understanding of detailed variability. The manuscript is well-written, although sometimes needing more objective comments and some polishing. "

We thank Reviewer 1 for the very complete and detailed review. We tried to address

all the comments, which helped to improve the manuscript. Please find below our response to each comment. The modifications (tracked in red) are in the attached manuscript.

"There is a strong focus on the low cost of the instrumentation, but that is really not very novel, since this type of devices have been developed and applied by numerous teams over the last two decades. However, they are becoming increasingly cheaper due to the reduction of hardware costs, development and miniaturization of sensors and datalogger and higher availability of open-source software. I therefore suggest that the authors put more focus on the novelty of the concept of moving the probes along the measuring site, rather than on the monitoring system itself, especially since this approach (monitoring) has not been conducted in this case-study."

We modified several sentences in the manuscript to emphasize that the main application of the Distributed Temperature Profiling (DTP) system in this study is moving sequentially the probes across the landscape to characterize permafrost distribution (e.g., line 17 on page 1 (in abstract), line 3 on page 5). However, since building low-cost (meaning duplicable), vertically resolved temperature probes is critical for sequential measurement, we retained the description of the cost parameters in the manuscript.

"Still on the measuring method (profiling), this type of approach was frequently used on local and urban climate studies, at least until the 1990's, for measuring air temperatures across large areas, with sampling measures at specific sites, which were then temperature-corrected for time, in order to allow for comparison. Such a correction might even be interesting to be done here, to make use of the near surface temperatures, which are affected by the diurnal changes (>25 _C following your manuscript). Examples of such application should be mentioned, since they link to the proposed method, that can be seen as precursor of the method. Here are some examples, but others also exist: https://doi.org/10.4113/jom.2010.1112, https://doi.org/10.1007/s00704-005-0152-1."

**TCD**

Thank you for this interesting suggestion. We cited the work of Hubbart et al. (2005), which is related to the measurement of air temperature at numerous locations (at the same time) to improve the understanding of factors influencing air temperature and possibly downscaling weather forcing. Hubbart et al. (2005) is a good example of many studies on this subject, including the ones provided by the reviewer (Alcoforado and Audrade (2004)). We added this reference to the manuscript (line 9 on page 3). However, it would be difficult to implement a time-space correction of temperature in this study as indicated by the reviewer because we do not have all the data required to do this (only two probes were kept at the same place during the acquisition), and because our focus is on soil temperature deeper than 0.25 m depth. We agree that keeping some DTP probes at static locations while moving others could have enabled a more in-depth analysis of near-surface temperature data (top 0.25 m). We will consider this for future data acquisition. In addition, we want to clarify that the change in near-surface temperature during our acquisition is not 25 oC as suggested by the reviewer. Between 9 am and 9 pm, the fluctuation in temperature was less than 5 oC at 0 cm depth, 3 oC at 8 cm depth and 1 oC at 16 cm depth. We evaluated these variations using a few probes that were kept at the same place between 9 am and 9 pm (this during different days in July), as well as using conventional monitoring locations. The spatial variability is generally much higher (see figure 2) than the observed temporal variability. This is why we decided to keep all the measurements in the figures. We have now clarified this observation in the paper (page 8).

"Temperature profiling has also been done by other authors on permafrost settings. For example, Goyanes et al (2014) (http://dx.doi.org/10.1016/j.geomorph.2014.04.010), measured temperatures from 5 to 70 cm depth in a grid in Deception Island to detect the effect of geothermal anomalies on permafrost distribution. They have also compared the results with ERT surveying. Although not with dataloggers and with a scarcer number of temperature measuring depths, but the approach is comparable to the one presented here, especially since in this manuscript the datalogging has not really been used. So, please check references and extend the review also to CALM

related publications, since I would think this has been done elsewhere."

Thanks for bringing this study to our attention. We added Goyanes et al (2014) to our references. We also discuss CALM related work including Nelson et al. (1998) and Shiklomanov et al. (2008) (line 14 on page 3).

"As you will find in the detailed review below, I think that you need to improve the characterization of the site in order to better assess the results. I think a larger-scale map derived from the UAV survey (or high resolution satellite) would be helpful, and also a window showing the setting of the studied slope in the watershed context. A discussion on the spatial variability of soil characteristics is needed, especially since the site is not homogeneous (as you mention, there are possible rock outcrops). Following this rationale, a description of the geomorphic units is also lacking, since concavities and convexities, may possibly be explained by different dynamics and also reflect soil (deposit?) types and hence will have an effect on soil temperature."

We added a larger map of the site with the watershed delineated (Figure 1a). We also provided additional information on the soil characteristics, geology and geomorphology, although such information is still limited at the site (line 25 on page 5). Finally, we clarified where the convexities are present in the studied domain (line 3 on page 12). We confirm that convexities (referred as topographic highs in the manuscript) are visible where near-surface permafrost is present (Figure 3).

"I also think you should be more cautious in what concerns to permafrost distribution, since it seems that you present no single direct observation of permafrost, other than the indirect measures by ERT. Please clarify this and indicate if there are other observations that show the characteristics of permafrost at the site or in its vicinity."

We are planning a future drilling campaign to collect direct information about the presence of permafrost, which continue to be hindered by logistical and permitting processes. We consider that soil temperature is a semi-direct observation of near-surface permafrost but we are particularly cautious on interpreting locations where permafrost

is deep or absent due to the current absence of deep borehole at the site (line 28 on page 17). In general, we believe that we interpreted the data with caution. We tried to improve the manuscript to clarify that our interpretation of the presence of permafrost in locations other than near-surface permafrost has much larger uncertainty (e.g., line 1 on page 13).

"As a conclusion, I think the manuscript is of good quality and should be published after a thorough review. The results show that the approach can be of wider application, especially with multiple datalogging systems installed, but the main-added value related to the way you apply the surveying."

Thanks for your review, including suggestions to improve this manuscript and extensions to other applications.

"Detailed comments: Title: I think the title is probably too ambitious for the contents. I would suggest a more focused title, such as "A distributed temperature profiling method for assessing spatial variability of ground temperatures (Nome, Alaska)". "

Based on this suggestion, we have modified the title to 'A distributed temperature profiling method for assessing spatial variability of ground temperatures in a discontinuous permafrost region of Alaska'

"Page 1, l. 10. Please clarify/rephrase the sentence, since temperature does not contain information on the properties modulating the soil thermal flux; it rather reflects these properties."

Time-series of temperature do contain information on properties modulating the soil thermal flux. For example, time-series can be used with inverse modelling technique to estimate soil thermal parameters (e.g., Tran et al., 2017, Nicolsky et al., 2009). We tried to improve clarity by modifying the sentence with "... as well as reflecting how various properties modulate the soil thermal flux" (line 10 on page 1)

"Page 1, l. 18. Replace AK by Alaska. The acronym is poorly known outside the US."

Done.

"Page 3, l. 25. I would suggest deleting the mention to Arctic permafrost regimes and write only "permafrost regimes", and add a few citations from non-Arctic regions (e.g. mountain and Antarctic)."

We improved this sentence accordingly to the reviewer's comment (line 32 on page 3).

"Page 4, l. 6. I reccomend extending this review to non-Arctic permafrost sites, since significant advances have been presented, for example, in the European Alps. "

We agree that advances have been made in many different environments, and we have included several studies outside the Arctic environment that focused on the use of temperature measurements (e.g., line 17 on page 3).

"Page 4, l. 19-20. Consider rephrasing the sentence, since it doesn't make too much sense to distinguish between soil minerals themselves and the organic component that may be mixed with the soil minerals, since you are aiming here to monitor the soil as a whole. "

We rewrote this sentence the following way: "Indeed, the energy exchange at the ground surface and in the subsurface is strongly mediated by snow, surface water, vegetation, and soil thermal properties, including peat layer thickness" (line 28 on page 4).

"Page 4, l. 35 – delete "low-cost". It has been mentioned earlier. "

Done.

"Page 5, l. 7 – Replace AK by Alaska. "

Done.

"Page 5, line 8 – Consider replacing "This"Teller watershed" by "The studied watershed along Teller Road" "

Done.

"Page 5, line 16-17 – Delete cumulative twice; it is not necessary, since annual precipitation data is always cumulative. "

Done.

"Page 5, lines 20-23 – You should rephrase and clarify if the most frequent situation is a thaw depth greater than 1 m, or a thaw depth of 0.4-0.6m. It is not clear (at least for the reader at this stage of the manuscript) why you are emphasising on such thaw depths, when there are also deeper ones. "

A thaw depth greater than 1 m is most frequent. We clarified the sentence (line 13 on page 6).

"Page 7, l. 1-5 Simplify text and make it more objective. Excessive use of adjectives associated to costs, etc. "

We reduced the use of low-cost throughout the manuscript.

"Page 7, line 17 . Where were those probes installed and what are their characteristics? "

The probes were not all pushed into the ground to the exact same depth due to permafrost or rock. These locations have required some interpolation. We clarified this (line 3 on page 8).

"Page 7, line 15-20. I find it difficult to accept that a system that was specifically designed to be tested for measuring soil temperatures, especially using such minute and fragile sensors, has not been calibrated. It is also not clear why temperatures at 80 cm depth were used, since not all sites were measured down to that depth. How many sites were extrapolated and what is the quality of the extrapolation? The assumptions that "no sensor-specific calibration curve could easily be defined while ensuring that it would increase sensor accuracy significantly" needs a better framing. If the system is

to be applied in the future, calibration is needed and mentioning that it could not have been done because it is 80 cm long, reveals some limitations for future developments. "

We improved the discussion to clarify how we performed the calibration assessment and why we did not apply additional calibration (line 14 on page 8). In short, we verified the manufacturer calibration and we assessed that the manufacturer calibration and its error assessment was better than what they described in the sensor reference manual. We did encounter some difficulties to calibrate the sensor to reach an accuracy better than 0.2 oC due to their limited resolution (0.06 oC), and the probe geometry (80 cm long) that implies large volume of calibration bath with temperature heterogeneity. We clarified this on line 14 on page 8. We agree there are some space for improvements in our calibration approach. We are currently developing much more effective calibration method. In addition, accuracy and resolution of digital sensor is improving constantly while digital sensors come factory-calibrated and are less sensitive to Electromagnetic Induced noise (vs analog, where you need to calibrate your ADC function).

"Page 8, l. 1 – Please clarify the setup. Is the pvc tube buried vertically with the sensors at different depths? What is the thickness of the tube? "

We clarified this in the manuscript (line 24 on page 8). It was a PVC rod with 5 mm diameter installed vertically with sensors attached to it by electrical tape.

"Page 8, line 26 – indicate where in the Arctic, since conditions are variable. "

Done (line 19 on page 9).

"Page 8, l. 31 – replace "topographic trends" by "topography". "

Done (line 24 on page 9).

"Page 9, l. 1 – replace "inferred" by "prepared". "

We replace 'inferred' by "reconstructed" (line 25 on page 9).

"Page 9, line 3. Delete "More details on UAV-based imaging..." and keep only the reference to Dafflon et al 2016. "

We improved the sentence (line 27 on page 9).

"Page 9, l.9-11 – delete. Not necessary. "

Done.

"Page 9, l.-13-15 – This should be moved to the methods. It answers the question raised above."

We moved it to page 8.

"Page 11 l. 6 – You mention vegetation and topography, but for such a detailed spatial and vertical analysis, you will certainly need to characterize geomorphology and soil type to better understand spatial variability. Add these characteristics to the study site description. "

We added some information on the geomorphology, soil properties and geology at the site (see response to major comment).

"Page 12, l. 2-5 – consider revising: "quite perpendicularly to the general slope aspect...". Improve phrasing "...possibly related to ground erosion and/or ground settlement". Why? This should be better discussed under the presentation of the study site. "

We improved clarity.

"Page 12, Line 12-13 – "...at many other locations for each depth..." - Clarify. "

We improved the sentence.

"Page 12, line 14 – "with a few exceptions for the shallowest measurement..." – clarify. "

We improved the sentence.

"Page 12, line 24-25 – Clarify what you mean with this sentence. "

We improved the sentence (line 14 on page 12, and line 2 on page 13). In particular we tried to clarify that based on monitoring data, it is obvious that the DTP dataset acquired on July 17 is not threatened by potential misinterpretation of near-surface permafrost where a seasonally frozen layer over a thick (> 1 m) perennially unfrozen soil is present.

"Page 14, line 16 – How can you associate directly the presence of permafrost with temperatures at of below 0.2 _C in mid-july? Please rephrase the sentence carefully, since you are using ERT data as a proxy for permafrost, which should work fine, but check the phrasing. "

We improved the sentence (line 25 on page 12). Comparison with the monitoring sites shows that if a frozen layer is present at the site, such a layer will be entirely thawed after June 15. As such, we think that soil temperatures at or close to 0oC is a strong indicator of near-surface permafrost.

"Page 14, line 20 – I would prefer indicating that the temperature values "suggest" the presence of permafrost, rather than "indicate", since you have no direct observations of permafrost along the transects. "

Done.

"Page 15, lines 14-15. Why does this happen? A scatterplot or at least R2 values could help understand the described co-variability, since the figure is really small. "

We did not show R2 values because all square values are relatively small (<0.5). We clarified this in the manuscript "Finally, it is to note that although lateral changes are observed at relatively similar locations in the various properties presented in this study (Figure 3, 5 and 6), the correlation coefficient for each possible combination of properties across the site is always smaller than 0.5" (line 15-16 on page 15). This means that the system is complex and is relatively clustered into various behaviors.

[Figure]

"Page 18, line 22 – Replace "temperature flux" by "heat flux". "

Done.

"Page 19, line 9 – Please rephrase the sentence reconsidering the novelty. "

We improved this sentence.

"Page 19, line 32 – You mention remote sensing data, but I think you mean satellite data, since with UAV surveys one can very well characterize structure of shrubs (multispectral, LIDAR and low elevation flights at multiple angle surveys)."

We mean satellite, UAV and aircraft surveys. The difficulty is to process this information correctly and disentangle various spatial scales; some related to microtopographic highs and lows, some others to drainage paths, some to rims along the slope gradient, and some to the slope. We modified our sentence to be more clear (line 32 on page 19).

"Figure 1 – Consider beginning the caption with a title that applies to both windows (i.e. Location and general setting of the study area) and only after mention "a) Aerial view...". Enlarge the dots of the two stations which are very difficult to depict in the image. Add graphic scale in Figure A. Although the UTM coordinates are there, they may not be self-explainable to many readers. A larger map of the UAV survey with topography would be important in order to assess the topographical and geomorphological setting of the slope section under analysis. That is very important to frame possible subsurface flow paths, geomorphic units and processes that may be active and which influence soil properties. "

Done (Figure 1). Also, see response to major comment.

"Figure 2. Add graphic scale. In caption, remove the mention to being overlain in figure 1. "

Done.

"Figure 3 – The permafrost areas identified with the rectangle are based in which data? The date is still not the one of maximum thaw depth, I suppose. Please clarify in the caption and in the text. The Y-axis for TDR is difficult to read. Check colour and font size. You mention that temperature above 25 cm were influenced bydiurnal oscillations. Hence, I think they should not be plotted. "

We improve figures, in particular the color. Font size was improved when possible while keeping Figure 3 on one page. We decided to keep temperature in the top 25 cm as they are not influenced in a critical way (see response to major comment). The temporal variability in temperature is much less than the spatial variability, even at the round surface. Thus, it makes sense to keep shallow temperature measurement. We improved the manuscript to clarify that uncertainty in comparing various locations is higher when close to the surface (line 3 on page 8).

"Figures 4 and 5 – If only the temperatures deeper than 25 cm should be used for comparison, I would suggest removing them from the figures."

See answer to previous comment.

Please also note the supplement to this comment:
https://www.the-cryosphere-discuss.net/tc-2018-264/tc-2018-264-AC1-supplement.pdf

———————————————————

[Figure]

**Supplement:**

[revised manuscript text omitted]

---

## Author Comment (AC2) · 25 Jul 2019

"The paper entitled 'Distributed Temperature Profiling System Provides Spatially Dense Measurements and Insights about Permafrost Distribution in an Arctic Watershed' by Léger et al. provides a highly interesting strategy for obtaining spatially and temporally dense soil temperature measurements at flexible spatial scales. The strategy is tested and evaluated in an Arctic watershed near Nome in Alaska. This is a well documented and thorough study that contains valuable results. The methodology is sound and the assumptions are clearly identified. It is a good paper, which also brought some new

data and knowledge on permafrost behavior in Alaska. Publication of this kind of paper will be very timely and beneficial for researchers working in the same field, as well as for many other researchers conducting a wide spectrum of environmental studies. The paper needs only minor revision, mainly regarding site description and readability of the figures. "

We thank this reviewer for the very helpful comments, which we have fully addressed in the revised manuscript. Please find below our response to each comment. The modifications (tracked in red) are in the attached manuscript.

"P2, L17 include "soil" Âż characterizing and monitoring soil temperature predominantly. . . "

We agree. Done.

"P2, L21 UTL-3 Scientific Datalogger (https://www.geotest.ch/en/expertise/measuringmonitoring/utl-3-temperature-datalogger.html) is frequently used in Europe and could be included in the list of loggers "

We added the name of this sensor when providing examples of temperature sensor/loggers and we referenced one study using it (line 24 on page 2).

"P2, L18-26 Consider to include one sentences briefly describing analog and digital temperature acquisition systems for ground temperature monitoring "

While historically analog measurements had the highest accuracy, current improvements in digital sensors will enable to reach similar accuracy. We added one sentence to clarify this (line 22 on page 2).

"P5, L13-14 and in presentation of results/discussion: I miss specific details related to soil type(s) at the study site. This is important in relation to the spatial variability of soil temperatures and corresponding interpretations. Some kind of information, e.g. soil profiles/variability in soil characteristics at the DTP sites could also be useful. "

We added more details on page 5.

"P9, L15 Suggest to replace "coldest temperatures" with "lowest temperatures". In also other parts of the manuscript (e.g P12, L2; P12, L13) "cold/warm temperatures" are used. Better write "low/high temperatures". "

Done.

"Figures: The readability and quality of the figures should be improved. Please read carefully the guidelines regarding figure composition, figure content and figure captions provided by TC at https://www.thecryosphere.net/for_authors/manuscript_preparation.html"

Thank you for this link. We improved Figures 1, 3, 5 and 6. Some of the fonts on Figure 3 are still relatively small to keep it on one single page, but we hope it is acceptable.

Please also note the supplement to this comment:
https://www.the-cryosphere-discuss.net/tc-2018-264/tc-2018-264-AC2-supplement.pdf